# Analytical Performance of NGS-Based Molecular Genetic Tests Used in the Diagnostic Workflow of Pheochromocytoma/Paraganglioma

**DOI:** 10.3390/cancers13164219

**Published:** 2021-08-22

**Authors:** Balazs Sarkadi, Istvan Liko, Gabor Nyiro, Peter Igaz, Henriett Butz, Attila Patocs

**Affiliations:** 1MTA-SE Hereditary Tumors Research Group, Eotvos Lorand Research Network, H-1089 Budapest, Hungary; sharkadi@gmail.com (B.S.); istvanliko@gmail.com (I.L.); butz.henriett@med.semmelweis-univ.hu (H.B.); 2Bionics Innovation Center, H-1089 Budapest, Hungary; nyirogabor1@gmail.com; 3MTA-SE Molecular Medicine Research Group, Eotvos Lorand Research Network, H-1083 Budapest, Hungary; igaz.peter@med.semmelweis-univ.hu; 4Department of Endocrinology, Department of Internal Medicine and Oncology, Semmelweis University, H-1083 Budapest, Hungary; 5Department of Laboratory Medicine, Semmelweis University, H-1089 Budapest, Hungary; 6Department of Molecular Genetics, National Institute of Oncology, H-1122 Budapest, Hungary

**Keywords:** Next Generation Sequencing, pheochromocytoma, paraganglioma, hereditary cancer, endocrine tumor syndrome

## Abstract

**Simple Summary:**

The escalating use of Next Generation Sequencing in the routine clinical setting greatly facilitates the genetic diagnosis of hereditary cancer syndromes. However, these novel methods pose new and unique challenges. In our study we sought to demonstrate the evolution of these techniques, especially whole exome sequencing and targeted panel sequencing. This study highlights the multi-layered workflow and how each step affects the diagnostic outcome and demonstrates the effectiveness of an in-house developed targeted panel sequencing for hereditary endocrine tumor syndromes.

**Abstract:**

Next Generation Sequencing (NGS)-based methods are high-throughput and cost-effective molecular genetic diagnostic tools. Targeted gene panel and whole exome sequencing (WES) are applied in clinical practice for assessing mutations of pheochromocytoma/paraganglioma (PPGL) associated genes, but the best strategy is debated. Germline mutations of at the least 18 PPGL genes are present in approximately 20–40% of patients, thus molecular genetic testing is recommended in all cases. We aimed to evaluate the analytical and clinical performances of NGS methods for mutation detection of PPGL-associated genes. WES (three different library preparation and bioinformatics workflows) and an in-house, hybridization based gene panel (**end**ocrine-**o**nco-**gen**e-panel- ENDOGENE) was evaluated on 37 (20 WES and 17 ENDOGENE) samples with known variants. After optimization of the bioinformatic workflow, 61 additional samples were tested prospectively. All clinically relevant variants were validated with Sanger sequencing. Target capture of PPGL genes differed markedly between WES platforms and genes tested. All known variants were correctly identified by all methods, but methods of library preparations, sequencing platforms and bioinformatical settings significantly affected the diagnostic accuracy. The ENDOGENE panel identified several pathogenic mutations and unusual genotype–phenotype associations suggesting that the whole panel should be used for identification of genetic susceptibility of PPGL.

## 1. Introduction 

Pheochromocytomas and paragangliomas (PPGL) are rare chromaffin cell tumors arising from the adrenal medulla or the sympathetic or parasympathetic paraganglia. PPGL have strong genetic determinism, overall approximately 40% of patients carry a germline mutation that predispose to the disease. The majority of these germline mutations occur in *SDHB*, *SDHD*, *VHL*, *NF1*, *RET* and *KIF1B* genes, but in rare or extremely rare cases, germline mutations of *SDHA*, *SDHAF2*, *EGLN1*, DLST, *FH*, *MAX*, *MDH2*, *KMT2D*, *TMEM127*, *MERTK*, *MET* and *SLC25A11* genes [1,2,3,4,5,6,7,8,9,10,11,12,13,14,15,16,17,18,19,20,21,22,23]. Moreover, several somatic driver mutations of *EPAS1*, *ATRX*, *IDH1*, M*ET*, *BRAF*, *HRAS*, and *FGFR1* genes have also been identified which may serve as target for specific therapeutical approaches as causative factors of the tumor [24,25].

It is recommended to perform genetic testing for certain groups at high risk for hereditary PPGL syndromes, which consists of positive family history, coexistence of multiple syndromic features, early onset, multiple primary PPGL, malignancy, extra-adrenal location, or combination of these features [26]. According to the actual guideline, phenotype-related genetic screening is suggested [26]. However, not all mutations manifest with specific phenotype, and in some cases, due to the low number of documented patients, genotype–phenotype correlations are not yet established [1].

Next Generation Sequencing (NGS) methods are categorized as high-throughput techniques that allow the parallel sequencing of multiple (even million) samples covering numerous genes or even the whole exome/genome. The appropriate informatics background is obligatory for the operation of these systems. The spreading of these techniques revolutionized the genetic and the hereditary disease diagnostics and reformed the everyday clinical practice. Beside their advantages, these methods yielded novel obstacles to overcome: the distribution of NGS techniques required technological upgrades, new expertise and workflow to be developed. Alongside the clinical practitioner, laboratory staff, bioinformatics specialists and molecular biologists synchronized work is mandatory for the correct assessment of the results. The appropriate choice for use is of utmost important due to the sheer amount of data generated by the process [27]. The indication varies between different tumors, but the American Society of Oncology recommends that if the chance of carrying an oncogene germline mutation exceeds 10% the patient should undergo genetic testing of the predisposing cancer genes [28] and patients affected with PPGL with the overall ~40% heterogeneity certainly exceed this criterion. This recommendation is supported by the fact that at least 10% of patients with “low risk” cases may carry predisposing mutation [4]. Due to the high number of various genomic aberrations that could lead to developing PPGL, the molecular genetic diagnosis easily becomes costly and burdensome [29,30,31,32,33]. WES started to emerge as both a research and a diagnostic tool for PPGL in the recent past [2,34,35]. Exome sequencing identified novel PPGL susceptibility genes and novel genes are predicted to be identified in the future [16,17,18,19,20,21,36]. NGS technologies are capable of screening familial [37,38,39,40,41] and sporadic cases [20,34]. With these technologies, novel somatic mutations can be identified [20,42,43,44,45] and screening large cohorts of PPGL patients became available [20,37,39]. Moreover, WES contributed to the complex profiling of these tumors [22,46].

However, despite the gradual decrease of experimental costs, whole-exome sequencing is still only sporadically used in routine diagnostics as the costs remain relatively high. Due to the various designs available, it is urgent to make a consensus to determine the indispensable quality standards for both technical processing and the interpretation of the results [2]. Various guidelines have set the standards and goals of genomic screening with NGS [47,48,49,50,51,52,53,54,55,56].

As a national reference center for Hereditary Endocrine Tumor syndromes in Hungary and part of European Reference Network for Endocrine Diseases (ENDO-ERN) our laboratory performs the molecular genetic analysis of patients with hereditary endocrine tumors. The incidence of these syndromes is low, but in order to provide the genetic test result within an acceptable time, we decided to use Next Generation Sequencing in the routine molecular genetic diagnostic workflow. In this recent work, we summarize our experience with NGS based methods in molecular genetic testing of PPGL. WES along with an in-house targeted gene sequencing panel (ENDOGENE) was tested on 82 patients and the analytical performance was evaluated.

## 2. Materials and Methods

### 2.1. Patients and the Genetic Testing of the RET, VHL, SDHB, SDHC, SDHD and TMEM127 Genes Using Sanger Sequencing

A retrospective medical and laboratory record review was performed on all patients diagnosed with hereditary endocrine tumor syndrome including suspicion of hereditary pheochromocytoma and paraganglioma during the period 1998–2020 under care at Semmelweis University, Budapest, Hungary. Our center is a national reference and part of European Reference Network (ERN) expertise center for hereditary endocrine tumors. After genetic counseling and having obtained informed consent, all patients with PPGL underwent genetic testing for the *RET*, *VHL*, *SDHB*, *SDHC*, *SDHD*, *MAX* and *TMEM127* genes using conventional methods including PCR amplification followed by Sanger sequencing as previously reported [57]. Blood DNA was extracted using commercially available DNA extraction kits (DNA isolation from mammalian blood, Roche, or DNA isolation kit from blood, Qiagen LTD). Bidirectional DNA sequencing of all these genes and large deletion analysis of the *VHL*, *SDHB*, *SDHC* and *SDHD* and *TMEM127* genes were performed using multiplex ligation probe amplification [58]. Of these samples 20 were used for determination of analytical sensitivity of whole exome sequencing (WES) performed between 2015–2019. In 2015, an in-house NGS based gene panel (ENDOGENE panel) was developed and introduced into clinical practice. Fifteen samples were used for the validation of ENDOGENE panel and additional 61 patients were tested prospectively. The study was approved by the Hungarian National Public Health Center (NPHC: 41189-7/2018/EÜIG, 13 December 2018) and the Scientific and Research Committee of the Medical Research Council of Ministry of Health, Hungary (ETT-TUKEB 4457/2012/EKU).

### 2.2. Whole Exome Sequencing

Seven members of two families presenting PPGL and 13 unrelated patients affected with PPGL were selected from our database containing the clinical and laboratory data of 241 patients and relatives diagnosed and treated at the 2nd Department of Internal Medicine, Semmelweis University with clinical diagnosis of PPGL between 1998–2019. Twelve patients carried *SDHB* mutations, two *SDHD* mutations and one *SDHC* mutation (Table 1). Six patients had no mutation in *SDHB*, *SDHC*, *SDHD*, *VHL*, *RET*, *TMEM127* and *MAX* genes. A total number of 29 missense/nonsense variants were identified with Sanger sequencing in this cohort. These variants were used as positive references, while wild type sequences were considered as negative references during evaluation of analytical performances of WES.

WES was performed in all 20 samples; four samples from a family presenting *SDHB* mutation were prepared using Agilent 51 M SureSelect Biotinylated RNA Library kit, 12 unrelated samples, were prepared using BGI 59 Mb exome kit and four samples were prepared using Illumina’s Rapid Capture Exome library preparation kit. WES was performed at NGS certified provider BGI Hong Kong (for libraries prepared with Agilent and BGI kits) [59,60] and by Omega Biotech, USA (samples prepared by Rapid Capture Exome). Library preparation and sequencing strategies are summarized in Appendix A. 

For Illumina workflow, the bioinformatics analysis followed the routine Illumina pipeline. The adapter sequence was removed, and low-quality reads which had too many Ns and low base quality bases were discarded. Burrows–Wheeler Aligner (BWA) [61] was used for the alignment. Single Nucleotide Polymorphisms (SNPs) were determined by SOAPsnp, Small Insertion/Deletion (InDels) were detected by Samtools/Genome Analysis ToolKit (GATK) [62], Single Nucleotide Variants (SNVs) were analyzed by Varscan, CNVs were detected by ExomeCNV/Varscan [63,64]. ANNOVAR and GATK FUNCOTATOR was used for annotation [62,65].

For *Complete genomics workflow* the VCF files were received from the sequencing provider.

The minimum sequencing depth for Illumina workflow was 10 reads/allele (20x) while for Complete Genomics data this threshold was set to 5 reads/allele (10x). Using in-house scripts written in phyton, the outputs of VCF files obtained either by Illumina or Complete Genomics platforms were merged into a single database file. Mean depth of coverage of PPGL genes were calculated using samtools bedcov utility on the CDS regions of genes obtained from gencode.hg19_v29 annotation.

### 2.3. Developing the ENDOGENE Panel

In the first version of ENDOGENE Panel (version 1.0), the covered genes included the *EGLN1*, *EPAS1*, *FH*, *KIF1B*, *MAX*, *MEN1*, *NF1*, *RET*, *SDHA*, *SDHAF2*, *SDHB*, *SDHC*, *SDHD*, *TMEM127* and *VHL* genes. During the development of the panel novel PPGL susceptibility genes were identified, therefore, the second version (version 2.0) included the *GOT2*, *MDH2* and *SLC25A11* genes as well. For targeted library preparation, a hybridization-based Roche NimbleGene SeqCap technology was used. Probes were designed for every exon and ± 30 bp intronic sites. The micro format of the MiSeq Reagent kit was used for ENDOGEN Panel v1.0, whereas the nano format was used for ENDOGEN Panel v2.0 (Illumina Inc., Foster City, CA, USA). The sequencing was carried out in our laboratory on Illumina MiSeq sequencing device (Illumina Inc., Foster City CA, USA).

The sequencing data was assessed with GATK (Genome Analysis Toolkit) following Best Practice guides [66]. The adapter sequences were removed with Cutadapt software [67]. The raw FASTQ format data was aligned to the UCSC hg19 human reference genome with BWA [61]. The reads below quality score 30 were removed GATK HaplotypeCaller [68]. PCR duplicates were removed with Picard MarkDuplicates (http://broadinstitute.github.io/picard; 6 August 2021) software. The indel realignment and the recalibration of the quality score was carried out with GATK v2.5-2 [62,66,69]. High quality InDels were filtered by criteria: “QD < 2.0, ReadPosRankSum < −20.0. The minimum sequencing depth was 20 reads/allele (40x)

Variant annotation was carried out with FUNCOTATOR, SNPEFFECT, SIFT, ClinVar, Varsome and PolyPhen applications [62,70,71,72]. The prevalence and the clinical impact of the variants were assessed using data from dbSNP [73], the American Exome Project Variants Server (National Heart, Lung, and Blood Institute Exome Sequencing Project Exome Variant Server (http://evs.gs.washington.edu/EVS; 15 March 2021)), Hapmap [74], ClinVar, Varsome 1000Genomes [75], gnomad [76] and LOVD [77] databases. 

Variant calls were subject to the same filtering parameters, eliminating non-exonic (untranslated region: UTR), synonymous and common (>1% MAF from the 1000 genome project, the exome sequencing project, and the Exome Aggregation Consortium) variants, as well as variants categorized as benign using ACMG criteria (ACMG criteria and PolyPhen-2 score = benign and SIFT < 0.05). All variants were categorized by recommendation of the NGS study group in PPGL, too [25].

All pathogenic, likely-pathogenic or variants with unknown significance were validated by Sanger sequencing.

## 3. Results

### 3.1. Whole Exome Sequencing

WES was performed on a set of 20 germline DNA samples obtained from patients with PPGL. Of these patients, seven belonged to two kindreds (F1 and F2) with already known *SDHB* p.Cys196Gly and *SDHB* p.Cys196Arg mutations. 

### 3.2. Depth of coverage

For WES, the offered minimum mean depth of coverage per sample by the manufacturers was 100 reads. This coverage was achieved with all three library preparations and sequencing platforms. No significant differences were found in the number of unique reads, bases corresponding to targeted sequences and bases with no coverage (Appendix A). 

However, analyzing the depth of coverage of PPGL-associated genes, significant differences were observed between genes and platforms. The most under covered regions belonged to *SDHA*, *SDHC* and *SDHD* genes in Agilent library preparation (Figure 1, Appendix A).

### 3.3. Analytical Validation

For analytical validation, Sanger sequencing was performed of all exons of *SDHB*, *SDHC*, *SDHD*, *TMEM127* and *VHL* genes and exons 10,11,14–16 of *RET* gene in all samples sequenced by WES. The number of nucleotides covered by Sanger sequencing was 3569. 

The total number of heterozygous, non-synonymous variants in these genes in these 20 samples were 29 variants. WES correctly detected all of them using an optimization of filtering strategy. The genetic alterations of samples used for analytical testing of WES are summarized in Table 1.

### 3.4. Optimization of Bioinformatical Workflow, Role of Allelic Ratio

As PPGL-associated pathogenic variants are heterozygous in germline DNA, we used this criterion for optimization of our bioinformatical workflow. In our study, the term of deviation (expressed in %) stands for the difference in modulus from the ideal allele fraction range (AFR) for heterozygote calls. For a heterozygote call the ratio of wild type and mutated allele is 1. The AFR shows the deviation of certain heterozygote variant from this number expressed in percentage (i.e., for a heterozygote call with 15/20 reads/alleles the AFR would be 75%). During WES data filtering we observed that some heterozygote variants showed larger allelic ratio. Based on the AFR the sensitivity of workflows was evaluated. Samples sequenced with BGI library preparation an AFR between 27–73% was needed for the correct identification of all true non-synonymous variants. For Agilent workflow, an AFR between 44.8–55.2% whereas for Nextera kit a ratio between 45.4–54.6% was necessary in order to achieve 100% sensitivity (Table 2). 

### 3.5. Design of the ENDOGENE Panel v1.0

Due to the need of an in-house validated assay, we developed a hybridization-based library preparation method. ENDOGENE Panel v1.0 was capable of the simultaneous sequencing of *EGLN1*, *EPAS1*, *FH*, *KIF1B*, *MAX*, *MEN1*, *NF1*, *RET*, *SDHA*, *SDHB*, *SDHC*, *SDHD*, *SDHAF2*, *TMEM127* and *VHL* genes. A total number of 509 fragments covered the genes listed above. The complete sequence spanned 126,116 nucleotides. 

For analytical validation of the ENDOGENE Panel v1.0, 15 patients with 10 verified pathogenic mutations (2 *RET*, 5 *SDHB*, 2 *TMEM127* and 1 *VHL*) were included. The coverage of the analyzed genomic regions was above 20 reads per allele (total 40x). Variants of 3′ and 5′ UTRs, of intron regions, synonymous variants and variants with coverage < 10 reads were excluded from further analysis. In total 155 variants mapped to the coding regions. Of these variants, 41 were true positive while 114 were false positive. No false negative variants were detected. Fifteen of the false positive calls were due to a single *MEN1* variant. The *MEN1* p.T546A (rs2959656) was labeled as normal in the reference sequence used. The reference genome of the *MEN1* gene differs in the databases, therefore a special caution is needed during annotation of the *MEN1* variants.

In order to decrease the number of false positives, we applied a filter based on the allelic fraction range (AFR%) described above. Variants with a ratio less than 0.3 or higher than 0.7 were excluded. All the previously verified 12 pathogenic variants were correctly identified. Two variants of unknown significance (VUS) and 25 benign polymorphisms were found. Using this additional filter, the sensitivity of the ENDOGEN Panel v1.0 was 100%, accompanied with 99.1% specificity.

### 3.6. The Prospective Group of ENDOGENE Panel v1.0

The diagnostic use of the ENDOGENE Panel v1.0 was tested in the clinical setting on 24 samples which had no previous genetic diagnosis. Using the criteria detailed above, 62 variants were identified. In all cases, the already mentioned *MEN1* variant was called and categorized as false positive. Out of the 24 patients, 9 (37.5%) carried pathogenic variants (2 *SDHB*, 7 *NF1*) and in one patient a novel *VHL* variant, classified as VUS was detected (*VHL*: p.36_37insSGPEE) in a young patient presenting with carotid body paraganglioma. The remaining 28 variants were variants categorized as benign polymorphisms. It is worth noting that in a patient the panel sequencing identified two different *SDHB* mutations which were verified with Sanger sequencing: beside a p.R90 frameshift mutation, an *SDHB* p.T88I variant was found too. 

### 3.7. Upgrading the ENDOGENE Panel v1.0 to v2.0 

During the last three years novel genetic susceptibility loci have been identified for PPGL. Therefore, we had to upgrade our panel by including 3 additional (*GOT2*, *MDH2* and *SLC25A11*) genes. The same bioinformatical pipeline was used. The effectiveness of the ENDOGENE Panel v2.0 was tested on 37 patients with no previous genetic diagnosis. Pathogenic variants were identified in 10 patients (27%). Mutations in *SDHB* (three patients), *FH* (two patients), *NF1* (four patients), and *VHL* (one patient) genes were detected and confirmed with Sanger sequencing. Four variants were categorized as VUS; the *SDHC*: c.94A > G (p.Thr32Ala) was found alongside one pathogenic variant suggesting that this variant might be a benign or a likely-benign variant. The pathogenic role of the *MDH2*: c.365G>A (p.Arg122Gln) and the *SDHA*: c.837G>T (p.Met279Ile) should be further tested following recommendation provided by the NGS in PPGL consensus statement [25]. Confronting data about the pathogenicity of the *RET* c.2372A>T (p.Tyr791Phe) have been presented, the detailed phenotype of our case is presented in Discussion section.

In summary, of 61 prospectively tested cases 19 (31.1%) harbored pathogenic/likely pathogenic variants (all variants detected in our cohort are summarized in Table 3). Of these variants, eight could be considered as novel as they have not been reported in any database to the best of our knowledge (Table 4). All these variants were confirmed by Sanger sequencing. Five of these is classified as pathogenic or likely pathogenic (all of these variants are truncating variants). Three variants are classified as VUS. Two *SDHB* variants: SDHB(NM_003000.3):c.263C>T (p.Thr88Ile); SDHB(NM_003000.3):c.268C>G (p.Arg90Gly) occurred in a patient where another pathogenic or likely pathogenic variant was identified (Figure 2, Panel A). The distribution of sequencing reads containing these variants show that these variants occurred at the same chromosome, therefore they are all in cis. The third VUS was detected in a patient with NF1 syndrome (Case 24). This is a complex alteration which has been annotated differently by various tools. However, looking at the sequence, this variant would is named NF1(NM_001042492.2):c.5047_5053delinsGGAG(p.Asn1683_Ser1684_Trp1685delinsGlyGly) (Figure 2, Panel B). 

No large deletions or copy number alterations were detected in our cases. Multiplex ligation-dependent probe amplification assays were used for analysis of *VHL* (probemix P016), *SDHB*, *SDHC*, *SDHD*, *SDHAF1* and *SDHAF2* (probemix P226). 

## 4. Discussion

Next Generation Sequencing and especially the targeted sequencing of certain chromosome regions and genes became the prime focus in the clinical management and the research of PPGL [36,78,79,80,81]. Even though methods covering the whole exome or even the whole genome are available, the targeted sequencing of certain genes is preferred in the clinical setting due to their cost-effectiveness [82,83,84].

PPGLs accompany various hereditary tumor syndromes. The genetic counseling and screening of these patients and their family are essential. Life-long monitoring is also compulsory for asymptomatic individuals carrying a pathogenic variant in PPGL-related gene. Depending on the affected gene, the childhood or even the prenatal genetic screening could be recommended, especially in case of *FH* and *SDHB* mutations due to their often aggressive, malignant phenotype [17,85]. This recommendation for early screening is further supported by the fact that there is no reliable marker for the malignant potential [86]. Tumor metabolomics and detailed immunohistochemistry of SDHB, FH and GLS1 enzymes may provide help in the future [87].

Molecular genetic tests for PPGL are recommended by recent guidelines [25,26,88]. Based on clinic-pathological conditions, a successive testing of genes associating with PPGL is recommended [24], but currently the availability and cost effectiveness of NGS methods are attractive options. However, the analytical and the clinical validation of these methods is mandatory before applying them in the clinical setting. During a test development of an in-house sequencing method, both gene panel and WES should follow the recommendation of The European Society of Human Genetics and only genes with known genotype–phenotype correlations can be investigated for diagnostic purposes [52]. Following this recommendation, we tested three independent library preparations and two sequencing strategies for their performances in testing of PPGL associated genes. First, a critical parameter was the coverage of our target genes with WES methods. The minimal coverage is highly depends on library preparation and sequencing devices, so universal recommendation for the minimal coverage cannot be made. The differences in them are represented in the pipeline of the sequencing method. Low coverage could indicate false negative variants, therefore the declaration of the minimal coverage of certain laboratories is mandatory [48]. However, high coverage is neither optimal due to the increasing sequencing costs and it yields more false positive calls. In germline testing, a min. 30x coverage is recommended, and in our study the 40-reads (20 per allele) coverage was enough for the identification of all pathogenic variants after an optimization of bioinformatical analysis. The sequencing depth of all the three tested library preparation provided sufficient coverage of all PPGL associated genes, but for the Agilent 51M exome kit the coverage of *SDHA*, *SDHC* and *SDHD* genes was the lowest. This observation is in line with previously reported data showing an inadequate coverage for the majority of variants in seven genes including *SDHC* and *SDHD* [89]. Despite this disadvantage, our data confirmed that WES can be a suitable tool for molecular genetic testing of inherited diseases. Position-specific comparative analysis of disease-causing variants of PPGL genes identified through NGS panels demonstrated that exome sequencing with a validated bioinformatical pipeline can be used for clinical testing [90]. Therefore, targeted analysis of PPGL genes from WES data may be suitable for clinical diagnostic purposes.

Parallel with WES, we developed an in-house gene panel sequencing (ENDOGENE Panel) for cost effect analysis of PPGL-associated genes together with *MEN1* gene. As a reference center for Hereditary Endocrine Tumors our laboratory routinely tests patients with PPGL and hereditary endocrine cancer syndromes. The first version of ENDOGENE was designed in 2015 and it was capable for sequencing of 15 hereditary endocrine tumor syndrome candidate genes. In order to assess the analytical performance, we first tested the effectiveness of the panel on samples with known pathogenic mutations and genetic diagnosis. The ENDOGENE panel successfully identified all known pathogenic mutations. In case of genetically negative cases, the panel sequencing did not identify a pathogenic mutation either. The sensitivity of our test was 100% with a specificity of 99.1%. These parameters are in line with those requirements established for germline testing by the Food and Drug Administration [91]. 

After validation, we used the ENDOGENE panel for prospective analysis of all patients referred for genetic analysis. In total, we identified pathogenic mutations in 19 of 61 (31.1%) of patients tested prospectively, which is in line with data previously reported [22,92]. It is important to note that mutations detected in Hungarian population were unique, no “founder” mutations have been detected. Therefore, only the specific phenotypes may guide the clinician in choosing the most accurate genetic test, but the successive testing of genes related to the well-known hereditary tumor syndromes (MEN2, VHL, NF1 and paraganglioma syndromes) would lead to a long and burdensome process. Our data confirms that both NGS approaches (gene panel and WES) have similar diagnostic yield in PPGL. The diagnostic yield, however, varies by diseases [93,94], but in apparently sporadic PPGL patients the prevalence of germline mutations is around 20–40%. Currently there is no recommendation for using NGS in molecular genetic testing of PPGL. However, for rare diseases the gene panel testing is preferred over WES. Based on our experience for non-syndromic PPGL the choice between panel and exome sequencing can be traced back to the availability of NGS platforms and cost. The major advantages of exome sequencing over targeted NGS panel testing is the evaluation of all coding regions in the genome. As shown in our study, even within this short period we had to upgrade our panel sequencing strategy because of the newly discovered genes. Based on our study, WES coverage depth was adequate for detection for close to all pathogenic variants identified on targeted NGS panel testing, along with newly-discovered PPGL genes. In addition, the repeated analysis of WES data may further increase the diagnostic yield of exome sequencing. The turn around time (TAT) for providing the genetic test report is 4–5 weeks, which includes Sanger validation from a separate DNA sample isolated from the same patient and pre- and posttest genetic counselling. Our panel sequencing is performed usually 1–2 times/month, depending on the requested number of tests. Generally, batches of eight to 24 samples are sequenced. With this strategy the cost of sequencing per sample is approx. 250 EUR. Contrarily, with Sanger sequencing the cost of sequencing only the most recommended PPGL genes (basic set: *SDHB*, *SDHD*, *SDHC*, *VHL*, *TMEM127*, *RET* and *MAX*), the TAT would take 3 months and the cost would be more than the 1000 EUR/sample.

Our study resulted in discoveries of unusual genotype–phenotype associations (Table 3). A VUS *VHL* variant (NM_000551.4): c.123_137dupAGAGTCCGGCCCGGA (p.Ser43_Glu47dup) was identified in a patient with carotid body paraganglioma, which would have been missed or delayed significantly if the routine protocol had been applied. In this case, testing of *SDHx* genes is recommended as a first test, while testing of the *VHL* gene is recommended only in case of other specific manifestations of the disease or the presence of von Hippel–Lindau syndrome in the family [95]. Since neither criterion was present in our patient, the genetic diagnosis with Sanger sequencing would have been a long and burdensome process. Genetic testing of the index patient’s parents showed the absence of this variant suggesting that it occurred de novo in our case.

The ENDOGENE Panel was capable of identifying a complex genetic variation in the *SDHB* gene in one patient. The SDHB(NM_003000.3):c.263C>T (p.Thr88Ile), SDHB(NM_003000.3):c.268C>G (p.Arg90Gly), SDHB(NM_003000.3):c.271_273del (p.Arg91del) variants was detected in a 14-year-old patient presenting with a large (14 × 8 × 17.5 cm) intraabdominal mass at the right side spanning the midline. Multiple bone metastases were also detected. The patient underwent a surgical intervention but, due to bleeding and the localization of the tumor, complete surgical removal was not possible. The histological examination showed pheochromocytoma. Chemotherapy with cyclophosphamide, vincristine and dacarbazine (CVD) and after 10 month of radiotherapy with administration of ^131^I-MIBG. After three years, the patient is in remission. The bone lesions are without any change. The family history was negative for any malignant disorder. DNA sample was available only from the index patient’s mother, but none of the identified variants were present. The pathogenic role, based on multiple predictions is attributed to the SDHB(NM_003000.3):c.271_273del (p.Arg91del) variant whereas the two other variants are classified as VUSs. These alterations located close to the pathogenic variant. Looking at the mapped sequencing reads it is evident that all these variants are present in the same reads, while other reads are normal. These distributions suggest that this complex rearrangment affects one chromosome and all these variants are in cis. 

Mutations of *FH* gene are associated with hereditary leiomyomatosis, “fumarate hydratase deficient renal cell cancer (RCC)” (“FH-deficient RCC”) [96] and in a very few cases with PPGL [17]. We identified a novel variant (c.1256C>T (p.S419L)) in a patient with this phenotype and the pathogenicity of this variant was supported by the lack of staining of the tumor sample with FH antibody on immunohistochemistry [97].

The ENDOGENE Panel v1.0 identified the *RET* p.M918T (rs74799832) mutation in a 33-year-old male patient in whom the referring clinical diagnosis was a unilateral pheochromocytoma. This mutation associates with a severe MEN2B phenotype, usually causing medullary thyroid cancer (MTC) at a very young age [98]. Eight years earlier the patient had a total thyroidectomy and lymph node dissection due to MTC. At that time the most common *RET* mutations (exon 10 and exon 11) were tested in another laboratory and no *RET* mutation was identified. After our genetic test result clinical, biochemical and imaging studies revealed that his serum calcitonin level was still elevated and a mediastinal lymph node metatasis was detected with Positron Emission Tomography and Computed Tomography (PET-CT) Scans. Although the patient received chemotherapy and recently tyrosine kinase inhibitor therapy, further progression of the disease was observed. Other MEN2B related manifestations were not documented. This patient carries the *SDHD* p.H50R variant as well. Currently this variant is categorized as benign, but there are confronting results about its association with PPGL. Our data may support its benign role.

Beside pathogenic or likely pathogenic variants, numerous variants classified as with uncertain significance were identified. These VUSs present major challenges in clinical practice. Following the recommendations of The American College of Medical Genetics and Genomics and the European Society of Human Genetics, these variants should be reported and interpreted on the molecular genetic test report but taking a clinical action is not recommended [48,51,52]. During their interpretation, various factors such as their minor allelic frequency, in silico predictions for the protein function and other supplementary evaluations must be carried out. The *SDHD* p.G12S variant’s phenotype altering effect was previously studied by our working group [99]. Our results implied that this variant is significantly more frequent in MEN2 patients than in the healthy population. This variant occurred in a *NF1* and a *VHL* mutation carrier patients suggesting that this variant has a minor role in disease development. 

The classification of the *RET* c.2372A>T (p.Y791F) variant is also debated (note 1 January 2021 in ClinVar). This variant was identified in a 59-year-old patient presenting with unilateral adrenal pheochromocytoma. After genetic test result, routine clinical, biochemical and imaging studies were performed for MEN2 related manifestations. His serum calcitonin, serum calcium and parathormone levels were within the reference range and no thyroid abnormalities were observed on thyroid ultrasonography. There are data showing that this variant does not increase the susceptibility for MTC [100] and recently a functional study proved that this variant exerted no pathogenetic effect in vivo in mice [101]. Taken together these data we suggest that this variant can be considered as a variant with unknown significance.

The *MDH2* c.686G>A (p.Arg229Gln) and the *SDHA* c.837G>T (p.Met279Ile) variants are classified as VUS. Both were identified in patients presenting with unilateral, adrenal pheochromocytomas. In these cases, no other clinical manifestations were detected. The pathogenicity of these variants should be considered given their MAFs and the lack of other pathogen variants in these patients. In these cases, evaluation of the metabolic features together with expression of enzymes on protein level could clarify their pathogenic role. For interpretation of the clinical relevance of a rare VUS additional studies (somatic mutation analysis, functional assays) are needed. These VUS should be reported on molecular genetic test reports, but no clinical action can be made until their pathogenic role has been confirmed [25]. Therefore, in our cases, yearly medical examinations were carried out.

The VHL(NM_000551.4):c.576delA (p.Asn193MetfsTer9) variant was identified in young male patient (15 years old) presenting with hormonally active adrenal pheochromocytoma. After genetic test, his regular (yearly performed) clinical, biochemical, imaging and ophthalmological studies revealed no sign of other VHL-related manifestations. Genetic screening was performed in his parents and the same variant was detected in her clinically healthy mother (45 years old). Her screening for VHL-associated manifestation showed no VHL-related manifestations. 

Several novel variants were identified in patients presenting with typical signs of Neurofibromatosis type 1. Earlier, due to the size of the gene and the obvious clinical symptoms (skin alterations: café au lait spot, neurofibromas, Lisch nodules) the genetic analysis was not performed in these cases. However, as NF1 is an autosomal dominant disorder with significant alterations which associate with decreased life expectancy [102], early diagnosis and adequate interventions are indicated. Patient No. 28 represents a 33-year-old female patient presented 6 years ago with multiple neurofibromas. Clinical, hormone laboratory and imaging studies detected no other manifestations. After 7 years no progression and no new manifestation occurred

## 5. Conclusions

In summary, our research group developed a hybridization based targeted sequencing panel for hereditary endocrine syndromes. The ENDOGENE Panel effectively verified the previously known mutations and uncovered novel variants in patients without genetic diagnosis in a cost-effective way. Respecting the limitations of our panel, it can be simply expanded by novel genes in the future. In the case of targeted sequencing the most important value to reach is 100% sensitivity. As false positive variants can be excluded via Sanger sequencing, the false negative results pose the greatest threat to the patients and their families.

## Figures and Tables

**Figure 1 cancers-13-04219-f001:**
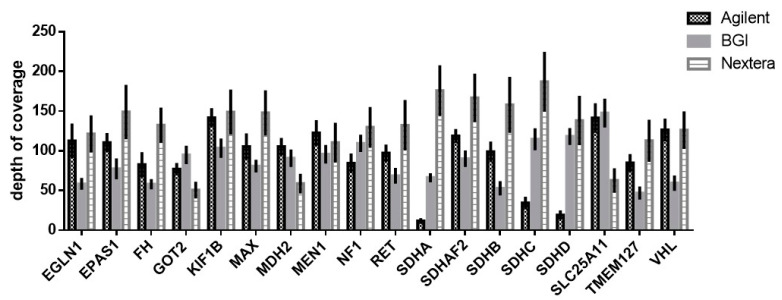
Coverage of PPGL associated genes by whole exome sequencing. Data is represented as mean ± SD.

**Figure 2 cancers-13-04219-f002:**
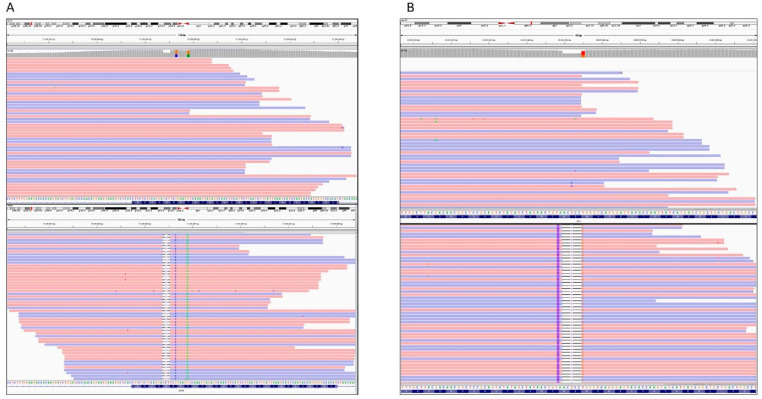
Schematic presentation of sequencing reads containing the variants SDHB(NM_003000.3):c.263C>T (p.Thr88Ile), SDHB(NM_003000.3):c.268C>G (p.Arg90Gly), SDHB(NM_003000.3):c.271_273del(p.Arg91del) (Case 38, (**A**)) and NF1(NM_001042492.2):c.5047_5053delinsGGAG(p.Asn1683_Ser1684_Trp1685delinsGlyGly) (Case 24, (**B**)). Each line represents one read. Half of the reads shows normal sequence (**upper part**) and half of the reads (**lower part**) show the mutated sequences.

**Table 1 cancers-13-04219-t001:** Genetic alterations of samples used for analytical testing of WES.

Patient ID	Known Mutation Detected by Sanger Sequencing	NGS Platform Used	Library Preparation Kit Used	Characteristics of Mutation Identified by Exome Sequencing		
				Mutation Confirmed	ACMG Category	Coverage, Read Number (Ratio and Read Numbers for Wild Type and Mutant Alleles)
1/F1	SDHB(NM_003000.3):c.586T>G (p.Cys196Gly)	Illumina Hiseq 2000	Agilent 51 M SureSelect	Yes	Pathogenic	50 (0.46: 27/23)
2/F1	SDHB(NM_003000.3):c.586T>G (p.Cys196Gly)			Yes	Pathogenic	58 (0.55: 26/32)
3/F1	No mutation detected			No mutation detected		
4/F1	No mutation detected			No mutation detected		
5	SDHB(NM_003000.2):c.649C>T (p.Arg217Cys)	Complete Genomics	BGI 59Mb Exome kit	Yes	Pathogenic	59 (0.38: 36/23)
6	SDHB(NM_003000.2):c.758G>A (p.Cys253Tyr)			Yes	Pathogenic	56 (0.59: 23/33)
7	SDHB(NM_003000.3):c.728G>A (p.Cy243Tyr)			Yes	Pathogenic	37 (0.62: 14/23)
8	SDHB(NM_003000.2):c.286+1G>A			Yes	Pathogenic	34 (0.5: 17/17)
9	SDHB(NM_003000.2): c.607G>T (p.Gly203 *)			Yes	Pathogenic	33 (0.36: 21/12)
10	*SDHC*(NM_003001.3):c.405+*1G*>*T*			Yes	Pathogenic	50 (0.42:29/21)
11	SDHD(NM_003002*.4*): c.147_148dupA (p.His50fs)			Yes	Pathogenic	33 (0.27: 24/9)
12	*SDHD*(NM_003002.4):*c*.*149A*>*G* (p.His50Arg)			Yes	VUS	36 (0.47: 19/17)
13	No mutation detected			No mutation detected		
14	No mutation detected			No mutation detected		
15	No mutation detected			No mutation detected		
16	No mutation detected			No mutation detected		
17	No mutation detected	Illumina Hiseq 2000	Rapid Capture Exome Library Preparation Kit	No mutation detected		
18/F2	SDHB(NM_003000.3):c.586T>C (p.Cys196Arg)			Yes	Pathogenic	185 (0.55:84/101)
19/F2	SDHB(NM_003000.3):c.586T>C (p.Cys196Arg)			Yes	Pathogenic	102 (0.48:53/49)
20/F2	SDHB(NM_003000.3):c.586T>C (p.Cys196Arg)			Yes	Pathogenic	170 (0.45:93/77)

* nomenclature.

**Table 2 cancers-13-04219-t002:** Analytical performances of WES for genes associated with PPGL using different cut-off values of allelic ratio for heterozygote calls.

	Agilent 51M SureSelect (*n* = 4)		Complete Genomics (*n* = 12)		Illumina Rapid Capture (*n* = 4)	
Allelic Ratio (%, Range)	30–70	41.1–58.8	30–70	41.1–58.8	30–70	41.1–58.8
True variants (mutations and polymorphisms) detected by Sanger sequencing	9		29		14	
Variants detected by WES	9	8	28	16	14	14
False positive variants	0	0	4	2	0	0
False negative variants	0	1	1	13	0	0
Sensitivity	100	88.9	96.5	55	100	100

**Table 3 cancers-13-04219-t003:** Variants identified with ENDOGEN panels v1.0, and v2.0 and the associated phenotypes.

ID	Panel	Phenotype	ACMG Classification		Clinical Classification Based on PPGL Consensus Guideline [25]
			Pathogenic/Likely Pathogenic Variants	VUS	
1	EP 1.0V	malignant PGL	SDHB(NM_003000.3):c.728G>A (p.Cys243Tyr)	-	pathogenic
2	EP 1.0V	Pheo	-	-	
3	EP 1.0V	malignant PGL	SDHB(NM_003000.3):c.586T>G (p.Cys196Gly)	-	pathogenic
4	EP 1.0V	malignant PGL	-	-	
5	EP 1.0V	Pheo	-	-	
6	EP 1.0V	MEN2	RET(NM_020975.6):c.1832G>A (p.Cys611Tyr)	-	pathogenic
7	EP 1.0V	Pheo	TMEM127(NM_001193304.3):c.419G>A (p.Cys140Tyr)	-	likely pathogenic
8	EP 1.0V	Pheo	-	-	
9	EP 1.0V	malignant PGL	SDHB(NM_003000.3):c.745T>C (p.Cys249Arg)	-	likely pathogenic
10	EP 1.0V	malignant PGL	SDHB(NM_003000.3):c.649C>T (p.Arg217Cys)	-	likely pathogenic
11	EP 1.0V	malignant PGL	SDHB(NM_003000.3):c.758G>A (p.Cys253Tyr)	-	pathogenic
12	EP 1.0V	MEN2B	RET(NM_020975.6):c.2753T>C (p.Met918Thr)		pathogenic
13	EP 1.0V	Pheo	TMEM127(NM_001193304.3):c.320delG (p.Ser107Ilefs*17)	-	likely pathogenic
14	EP 1.0V	VHL	VHL(NM_000551.4):c.407T>G (p. Phe136Cys)	-	likely pathogenic
15	EP 1.0V	Pheo	-	-	
16	EP 1.0P	Pheo	-	-	
17	EP 1.0P	Pheo	-	-	
18	EP 1.0P	Pheo	-	-	
19	EP 1.0P	malignant PGL	SDHB(NM_003000.3):c.286+2T>A	-	likely pathogenic
20	EP 1.0P	PGL	-	-	
21	EP 1.0P	NF1	NF1(NM_001042492.3):c.1756_1759delACTA (p.Thr586ValfsTer18)	-	pathogenic
22	EP 1.0P	NF1	NF1(NM_001042492.2):c.5047_5053delinsGGAG (p.Asn1683_Ser1684_Trp1685delinsGlyGly)	-	VUS
23	EP 1.0P	NF1	NF1(NM_001042492.2):c.4230_4231delCC (p.Leu1411GlnfsTer12)	-	likely pathogenic
24	EP 1.0P	NF1	NF1(NM_001042492.2):c.1466A>G (p.Tyr489Cys)	-	pathogenic
25	EP 1.0P	NF1	NF1(NM_001042492.2):c.2251+1G>A	-	likely pathogenic
26	EP 1.0P	NF1	NF1(NM_001042492.2):c.7465_7466insG (p.Lys2489ArgfsTer13)	-	likely pathogenic
27	EP 1.0P	Pheo	-	-	
28	EP 1.0P	NF1	NF1(NM_001042492.2):c.4175dupT (p.Val1393GlyfsTer2)	-	likely pathogenic
29	EP 1.0P	Pheo	-	-	
30	EP 1.0P	Pheo	-	-	
31	EP 1.0P	Pheo	-	-	
32	EP 1.0P	Pheo	-	-	
33	EP 1.0P	Pheo	-	-	
34	EP 1.0P	PGL-glomus caroticum	-	VHL(NM_000551.4):c.123_137dupAGAGTCCGGCCCGGA (p.Ser43_Glu47dup) = NM_000551.3(VHL):c.123_137dup (p.38_42SGPEE [3])	VUS
35	EP 1.0P	Pheo	-	-	
36	EP 1.0P	malignant PGL	SDHB(NM_003000.3):c.263C>T (p.Thr88Ile) SDHB(NM_003000.3):c.268C>G (p.Arg90Gly) SDHB(NM_003000.3):c.271_273del (p.Arg91del)	-	VUS VUS likely pathogenic
37	EP 1.0P	Pheo	-	-	
38	EP 1.0P	Pheo	-	-	
39	EP 1.0P	Pheo	-	-	
40	EP 2.0	Pheo	-	-	
41	EP 2.0	Pheo	SDHB(NM_003000.3):c.193C>T (p.Leu65Phe)	-	likely pathogenic
42	EP 2.0	Pheo	-	-	
43	EP 2.0	Pheo	-	-	
44	EP 2.0	Pheo	-	-	
45	EP 2.0	Pheo	-	-	
46	EP 2.0	Pheo	-	-	
47	EP 2.0	Pheo	-	-	
48	EP 2.0	Pheo	VHL(NM_000551.4):c.576delA (p.Asn193MetfsTer9)	-	likely pathogenic
49	EP 2.0	Pheo	-	-	
50	EP 2.0	Pheo&PGL	SDHB(NM_003000.3):c.286+2T>A		likely pathogenic
51	EP 2.0	Pheo	FH(NM_000143.4):c.1127A>C (p.Gln376Pro)	-	likely pathogenic
52	EP 2.0	Pheo	-	-	
53	EP 2.0	Pheo	-	-	
54	EP 2.0	abdominal PGL	SDHB(NM_003000.3):c.689G>A(p.Arg230His)	-	pathogenic
55	EP 2.0	Pheo	-	-	
56	EP 2.0	Pheo	-	-	
57	EP 2.0	malignant PGL	-		
58	EP 2.0	Pheo	-	-	
59	EP 2.0	Pheo	-	-	
60	EP 2.0	cervical PGL	-	-	
61	EP 2.0	NF1	NF1(NM_001042492.2):c.3456dupA (p.Leu1153ThrfsTer42)	-	pathogenic
62	EP 2.0	Pheo	-	-	
63	EP 2.0	NF1	NF1(NM_001042492.2):c.888+2T>G		pathogenic
64	EP 2.0	Pheo	-	-	
65	EP 2.0	Fumarase deficient leiomyoma	FH(NM_000143.4):c.1256C>T (p.Ser419Leu)	-	likely pathogenic
66	EP 2.0	Pheo	-	-	
67	EP 2.0	Pheo	-	MDH2(NM_005918.4):c.686G>A (p.Arg229Gln)	VUS
68	EP 2.0	Pheo	-	SDHA(NM_004168.4):c.837G>T (p.Met279Ile)	VUS
69	EP 2.0	Pheo	-	-	
70	EP 2.0	Pheo	-	-	
71	EP 2.0	Pheo		RET(NM_020975.6):c.2372A>T (p.Tyr791Phe)-	VUS
72	EP 2.0	NF1	NF1(NM_001042492.2):c.6850_6853delACTT (p.Tyr2285fs)	SDHC(NM_003001.5):c.94A>G (p.Thr32Ala)	The *NF1* variant pathogenic The *SDHC* variant VUS
73	EP 2.0	Pheo	-	-	
74	EP 2.0	Pheo	-	-	
75	EP 2.0	Pheo	-	-	
76	EP 2.0	NF1	NF1(NM_001042492.3):c.2991-1G>C	-	pathogenic

EP1.0V: ENDOGENE Panel version 1-validation group; EP1.0P: ENDOGENE Panel version 1-prospective group; Pheo: pheochromocytoma; PGL: paraganglioma; MEN: multiple endocrine neoplasia; NF1: Neurofibromatosis type 1; VUS: variant of uncertain significance. Patients tested by EP 1.0V had genetic diagnosis before panel sequencing. Patients tested by EP 1.0P and EP2.0 did not have a genetic diagnosis before testing.

**Table 4 cancers-13-04219-t004:** Novel genetic variants and associated clinical phenotypes identified in our recent cohort.

Sample ID	Manifestations	Age (Years)	Benign/Malignant	Genetic Variant	Clinical Significance
22	Neurofibromatosis Type 1: multiple neurofibromas Adrenal pheochromocytoma	30 30	B B	NF1(NM_001042492.2):c.5047_5053delinsGGAG(p.Asn1683_Ser1684_Trp1685delinsGlyGly)	VUS
23	Neurofibromatosis Type 1: multiple neurofibromas	32	B	NF1(NM_001042492.2):c.4230_4231delCC (p.Leu1411GlnfsTer12)	likely pathogenic
26	Neurofibromatosis Type 1: Adrenal pheochromocytoma	15	B	NF1(NM_001042492.2):c.7465_7466insG (p.Lys2489ArgfsTer13)	likely pathogenic
28	Neurofibromatosis Type 1	26	B	NF1(NM_001042492.2):c.4175dupT (p.Val1393GlyfsTer2)	likely pathogenic
36	Extra-adrenal PGL	14	M	SDHB(NM_003000.3):c.263C>T (p.Thr88Ile) SDHB(NM_003000.3):c.268C>G (p.Arg90Gly) SDHB(NM_003000.3):c.271_273del (p.Arg91del)	VUS VUS likely pathogenic
48	Adrenal pheochromocytoma	15	B	VHL(NM_000551.4):c.576delA (p.Asn193MetfsTer9)	likely pathogenic

## Data Availability

The data presented in this study are available on request from the corresponding author.

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
