# Peer review of "Analytical Performance of NGS-Based Molecular Genetic Tests Used in the Diagnostic Workflow of Pheochromocytoma/Paraganglioma"

_cancers, 2021, doi:10.3390/cancers13164219_

Round 1
Reviewer 1 Report
The authors have improved the manuscript substantially.
However, they have forgotten to delete some MEN1 mutations in table 3 (patient ID13 and ID14) :
In table 4, there is one patient with a MEN1 phenotype. The authors claim in the Materials and methods section that all the patients have PPGL. It will be more accurate to remove this patient.
Yes, hence our panel covers the MEN1 gene too, we included this patient in this list, but according to the Reviewer’s suggestion we removed this case from the revised version.
Author Response
We thank the reviewer for his/her comment. We removed the MEN1 patient from Table 4.
Reviewer 2 Report
Accepted
Author Response
Thank you for your positive feedback
This manuscript is a resubmission of an earlier submission. The following is a list of the peer review reports and author responses from that submission.
Round 1
Reviewer 1 Report
This is a study for nation wide Hungary regarding genetic screening for germline mutations by whole exon sequencing of about 15 pheochromocytoma-paraganglioma susceptibility genes including all genes in which germline mutations have been described in such patients. The need of WES is clearly explained. The methods are ok und of best quality and modern. Controls and subjects at risk are well selected without any source of overestimation of likelihood.
The results are clearly described.
I recommend: In those cases with pathogenic or likely pathogenic variants we should have information about pretest and posttest clinical data. For exemple: MENB case: any sklettal abnormalities? MTC metastases? VHL cases: completely screened for CNS, eyes, RCC, Pheo and pancreas? Which lesions? similarly for NF1, SDHB, C and D.
We should have an idea how much time WES in contrast to Sanger takes in this setting.
We should know something about costs of WES and Sanger.
Reviewer 2 Report
Sarkadi et al present an interesting study on the comparison between NGS based molecular testing and exome bases molecular testing in paraganglioma.
Major point:
- The comparison between the different technics is not clear in this study
- The mutation nomenclature in the different tables has to follow the HGVS guidelines
- Do the authors have confirmed the new mutations by Sanger sequencing? Notably for the multiple variants. If it wasn't done it is an important experiment to perform.
- No deletion or duplication is reported, which is weird. Do the analysis pipelines detect this genetic event?
- Why do the authors have chosen 10X of coverage? 30X is a more classic coverage.
- Why do the authors use the ACMG criteria and not the NGSinPPGL study group criteria for the variation classification?
- In the different table with variants, the report of likely benign or benign variation in the VUS column is confusing
- Some patients seem to have multiple variants in the same gene. Do these variants are in cis or in trans? How do the authors explain the occurrence of two variants in the same patient?
Minor point:
- In introduction page 2, the list of PGL susceptibility genes is not complete
- In introduction page 2, KIF1B gene mutations in PPGL are controversy, and a recent report questions their involvement in PPGL.
- In introduction page 2, the authors write that it is recommended to perform genetic test for high risk patients. It isn't what it is recommended by the Endocrine Society guidelines and by the European Society of hypertension guidelines.
- In table 1, patient 7, are you sure of the reported variant? p.Cys243Thre is not possible at this codon.
- In table 4, there is one patient with a MEN1 phenotype. The authors claim in the Materials and methods section that all the patients have PPGL. It will be more accurate to remove this patient.
- The classification of the EPAS1 variant p.S12L in likely pathogenic seems exaggerated: the variant isn't near one of the functional domain of the protein, and hasn't be described in the literature.
Reviewer 3 Report
The paper is presenting a newly designed NGS based tool for rare pheochromocytoma/paraganglioma cases. The data are interesting and convincing, chosen methods are accurate. In general deserves publication also because a rare occurrence of PPGL and related diagnostic burden.
However, paper requires some major improvements and changes
Line 274 p.8 “The GOT2: c.1178G>A (p.R393Q) and the RET c.2372A>T (p.Y791F) variants were classified as benign or likely benign variants”
The RET variant c.2372A>T is considered pathogenic (ACMG: PP5, PM1,PM2, ATA MEN2 Guidelines,)
Table 2 Title - “values of allelic ration” (ratio?)
Line 286 „MEN: multiplex endocrine neoplasia” – multiple?
A major revision should be made to mutation description in order to meet HGVS recommendations – as there is a lot of inconsistency:
p.6
SDHB: c286+1G/A - dot is missing
SDHB: p.G203Stop* - incorrect. Should be p.G203* (preferred because single letter for protein is in use ) or p.G203Ter, The p.G203STOP (or STP) should be avoided
and than next line:
SDHC: ivs+1G/T - a number for ivs is missing
p.8
SDHB: g.chr1:17349140C>T; c.728G>A (p.C243Y) (and other similar) – the “g” letter sometimes is used, sometimes not. It should be used together with reference number of genomic sequence, not chromosomal coordinates.
*: novel variants. – use another symbol. “*” must be used only for stop codon otherwise is confusing
SDHB: chr1:17359553A>T; c.e3+2T>A (splice mutation) – use IVS3 instead of e3, the same here: NF1: chr17:29509685T>G c.e8+2T>G
p.10
g.chr17:29550494_29550497delTAAC
c.1754_1757delTAAC (p.T586fs) or NF1(NM_001042492.3):c.1756_1759delACTA (p.Thr586ValfsTer18)
What is the justification for using “or” here?
"Wild type" is not recommended and avoided. Reference or just use "No mutation" instead in table
Please state clearly that canonical transcripts for coding sequence were used or indicate transcript accession like here:
NF1(NM_001042492.3):c.1756_1759delACTA (p.Thr586ValfsTer18)
Consider please if you using single or three-letter abbreviations for amino acids and use it consistently.
These are only selected examples of mistakes. Please revise thoroughly ALL of the mutation descriptions, and follow precisely HGVS recommendations (https://varnomen.hgvs.org/recommendations/DNA/) and use it consistently.